# Pattern of Antibiotic Use among Hospitalized Patients at a Level One Multidisciplinary Care Hospital

**DOI:** 10.3390/healthcare11091302

**Published:** 2023-05-03

**Authors:** Viviana Hodoșan, Lucia Georgeta Daina, Dana Carmen Zaha, Petru Cotrău, Adriana Vladu, Florica Ramona Dorobanțu, Marcel Ovidiu Negrău, Elena Emilia Babeș, Victor Vlad Babeș, Cristian Marius Daina

**Affiliations:** 1Faculty of Medicine and Pharmacy, Doctoral School of Biomedical Sciences, University of Oradea, 1st University Street, 410087 Oradea, Romania; 2Psycho-Neurosciences and Recovery Department, Faculty of Medicine and Pharmacy, University of Oradea, 1st University Street, 410087 Oradea, Romania; 3Department of Preclinical Disciplines, Faculty of Medicine and Pharmacy, University of Oradea, 1st University Street, 410087 Oradea, Romania; 4Department of Medical Disciplines, Faculty of Medicine and Pharmacy, University of Oradea, 1st University Street, 410087 Oradea, Romania; 5Department of Surgical Disciplines, Faculty of Medicine and Pharmacy, University of Oradea, 1st University Street, 410087 Oradea, Romania

**Keywords:** antibiotics, antimicrobial resistance, hospital, AWaRe, WHO

## Abstract

Background: Antimicrobial resistance is one of the world’s most serious health issues. Antibiotic resistance, excessive drug expense, and an increased risk of adverse reactions are all common outcomes of incorrect antibiotic prescribing. The goal of this study was to evaluate the prevalence of antibiotic prescriptions for inpatients to find areas for improvement. Methods: A retrospective study at Emergency Clinical County Hospital of Oradea, Romania was performed for five years between 2017 and 2021. Data was collected using medical records of the patients and reports from the pharmacy. Antibiotic consumption was expressed as DDD/100 BD according to the World Health Organization (WHO) by antibiotics, classes, and AWaRe classification. Results: The prevalence of antibiotic prescription was 53.8% during five years evaluated with a significant increase in 2021. A total of 13,677.42 DDD/100 BD antibiotics were prescribed, especially for surgical and medical prophylaxes. The most prescribed antibiotics were ceftriaxone, followed by metronidazole, and cefuroxime but there were some differences between years and wards. The most frequent antibiotic classes prescribed were cephalosporins (43.73%). The use of Watch Group antibiotics was high in all wards (59.69%). Conclusions: The prevalence of antibiotic use was high with cephalosporins being the most prescribed antibiotics. As a result, interventions are required.

## 1. Introduction

Antibiotics are the most prescribed drugs in hospitals, and their inappropriate use contributes to the rise in antimicrobial resistance (AMR) which is a global problem. Antibiotic use has increased in the past, putting selective pressure on susceptible bacteria, resulting in the growth of AMR. Antimicrobial resistance is one of the most serious public health issues, it cannot necessarily be completely eradicated but can be managed. As a result, attempts to decrease antimicrobial resistance by using suitable antibiotics are gaining universal attention [1]. Treatment failure increases medical costs and hospital stays, and *Clostridioides difficile* infection resulting from the misuse of antibiotics. Irrational prescribing can be result of a lack of drug understanding, unethical drug promotion, a high patient load, inefficient laboratory facilities, medication availability, ineffective law enforcement, and failure to assure compliance with recommendations [2,3]. 

The World Health Organization (WHO) published the first global surveillance report on antibiotic resistance in 2014 showing that six WHO regions had more than 50% resistance to third generation cephalosporins and fluoroquinolones in *Escherichia coli* and methicillin resistance in *Staphylococcus aureus,* and more than 50% resistance to third generation cephalosporins and carbapenems was reported in *Klebsiella pneumoniae* in hospital settings. The same report attributed 45% of deaths in both Africa and South-East Asia to multi-drug resistant (MDR) bacteria, especially *Klebsiella pneumoniae* [4].

There was an estimated average of 4.95 million deaths associated with bacterial AMR in 2019 [5]. All regions of the world are involved, but it is estimated the all-age death rate attributable to resistance is highest in western Sub-Saharan Africa (27.3 deaths per 100,000), and lowest in Australasia (6.5 deaths per 100,000). Lower respiratory infections accounted for more than 1.5 million deaths associated with AMR in 2019, making it the most prevalent infectious disease. There were six pathogens associated with resistance (*Escherichia coli*, followed by *Staphylococcus aureus*, *Klebsiella pneumoniae*, *Streptococcus pneumoniae*, *Acinetobacter baumannii*, and *Pseudomonas aeruginosa*) which were responsible for 3.57 million deaths associated with AMR in 2019 [5]. Meticillin-resistant *Staphylococcus aureus* caused more than 100,000 deaths attributable to AMR in 2019, while six more each caused 50,000–100,000 deaths: MDR excluding extensively drug-resistant tuberculosis, third generation cephalosporin-resistant *Escherichia coli*, carbapenem-resistant *Acinetobacter baumannii*, fluoroquinolone-resistant *Escherichia coli*, carbapenem-resistant *Klebsiella pneumoniae*, and third generation cephalosporin-resistant *Klebsiella pneumoniae* [5].

The key issue is that antibiotic use and a high number of prescriptions do not follow the optimum pattern [6]. Antimicrobial drug overuse has resulted in resistance in practically all antibiotic families and compared to the rate at which resistance is developing, new antibiotics for treating diseases are not being identified as quickly. As a result, it is critical to ensure that antibiotics are used appropriately through antibiotic stewardship programs (ASP) and perform annual qualitative analyses of antibiotic use. 

Previous studies have reported that 28 to 68% of antibiotic prescriptions in hospitals and ambulatory care facilities are inappropriate and that broad-spectrum antibiotics are overprescribed [7,8]. Every country already has a unique public healthcare system that severely prohibits reimbursement for improper medications. However, this technique is solely dependent on the study of the prescription-to-diagnostic code mismatch. As a result, the healthcare authorities have adopted some more innovative antibiotic use policies. Antibiotic use must be improved both quantitatively (lowering antibiotic administration time and usage) and qualitatively (surgical preventative antibiotic type, first administration time alteration) because of these efforts [9]. 

The Centers for Disease Control and Prevention (CDC) advised all hospitals to use ASP and outlined the basic principles of hospital-based ASP to optimize antibiotic use [10]. Some hospitals used these guidelines, and they observed significant reductions in antibiotic use, length of hospital stays, and *Clostridioides difficile* infection rates [11,12]. Potential interventions for ASPs include improving diagnostic accuracy and etiology based on culture results, optimizing the duration of treatment according to specific current guidelines, reducing extended antibiotic prophylaxis to prevent surgical site infections (SSIs), and encouraging de-escalation to oral antibiotics. Antimicrobial therapy is considered essential for the proper management of sepsis; early administration of effective antibiotics is lifesaving. ASP should work in a complex team composed of doctors, experts from the pharmacy, and microbiology laboratory to optimize the treatment of sepsis and other infectious diseases [13,14]. Important directions in this case are developing antibiotic recommendations for sepsis based on local microbiology data, elaborating protocols to administer antibiotics quickly in cases of suspected sepsis, reviewing antibiotic treatment, and stopping unnecessary antibiotics or de-escalation. However, it is quite difficult to implement ASP fundamental principles at a national level in many countries due to low clinician compliance, a lack of competence, and the lack of a suitable reward structure [15].

Antibiotic consumption can be expressed in grams, number of units, number of prescriptions or cost but these variables are different between hospitals, regions, and countries over time. This is why it is difficult to compare local and international consumption. To solve this issue, a technical unit of measurement named Defined Daily Dose (DDD) was created by the WHO International Working Group on Drug Statistics Methodology. DDD is the assumed average maintenance dose per day for a drug used for its main indication in adults, but DDD is not necessarily the same as Prescribed Daily Dose (PDD). It is obvious that therapeutic doses for patients often differ from DDD as they will be based on individual characteristics such as age, weight, type of disease, and many other pharmacokinetic considerations.

A very useful instrument to monitor the ASP is the AWaRe classification developed by the expert committee of the WHO using the Essential Medicine List (EML) in 2017 [16,17,18,19]. The AWaRe classification, updated from 2021, includes a total of 258 antibiotics classified into Access, Watch, Reserve and Not recommended groups by considering the impact of different antibiotics or classes on development of antimicrobial resistance and the importance of their appropriate use. The Access Group consists of antibiotics with the best therapeutic value at the same time minimizing the potential for development of resistance and they can be first or second choice for the 25 most common infections. In the Watch Group, included antibiotics are indicated for a specific number of critically infectious diseases, but they can be a target of antibiotic resistance and their prescriptions must be monitored. The Reserve Group is the last option when other antibiotics failed for highly selected patients (such as infections due to multi-drug-resistant bacteria), very carefully monitored and prioritized as targets of stewardship programs to ensure their continued effectiveness. According to the WHO 13th General Program of Work 2019–2023, at least 60% of total national antibiotic consumption must be Access Group antibiotics.

The aim of this study was to undertake an antibiotic prescription review to identify the possible measures and interventions for promoting a rationale use of antibiotics by using the WHO AWaRe classification and to plan future antimicrobial stewardship efforts. 

## 2. Materials and Methods

### 2.1. Study Design

A retrospective investigation was conducted at the Emergency Clinical County Hospital of Oradea, Romania for five years. Our teaching hospital has a tertiary one with a capacity of >500 beds and a large-scale of disease. It includes Intensive Care Units (ICUs), surgical and medical wards, (gynecology, plastic surgery, oncology, hematology, orthopedics, neurosurgery, ophthalmology, cardiology, and pediatrics) as main departments. Oral administration routes and intravenous, intramuscular injections were included. Antifungals, antiviral, anti-tuberculosis, and anti-parasitic or nebulization drugs were excluded.

### 2.2. Collection of Data and Calculation

Data regarding all antibiotic administration and diagnoses were collected from the hospital’s software program InfoWorld that electronically stores patient files, explorations, and treatment information. There are recorded antibiotic names, doses, routes of administration, and departments. 

Data on antimicrobial use between January 2017 and December 2021 were extracted from the pharmacy information system including consumption of intravenous and oral antibiotics. An evaluation was conducted for antibiotics prescribed according to the WHO Collaborating Centre for Drug Statistics Methodology, Anatomical Therapeutic Chemical (ATC)/DDD Index 2022 [20]. Antibiotic use was quantified in grams of each antimicrobial used and the result was divided by the WHO-assigned DDD resulting number of DDD (DDDs). Antimicrobial use density was expressed as DDDs/100 bed days (BD), for each antibiotic, classes, and route of administration. 

### 2.3. Classification of Antibiotics

We classified antibiotic agents into beta-lactams, macrolides, glycopeptides, lincosamide, polymyxins, tetracyclines, oxazolidinone fluoroquinolones, aminoglycosides, rifamycin, and azoles. Fosfomycin for oral administration was included in the category of other antibiotics for systemic use. The beta-lactams evaluated were penicillins, combination beta-lactam/beta-lactamase inhibitors, and cephalosporins/beta-lactamase inhibitors, 1st-generation cephalosporins, 2nd-generation cephalosporins, 3rd-generation cephalosporins, 4th-generation cephalosporins, and carbapenems. We performed an analysis of the distribution of antibiotics using the 2021 WHO AWaRe classification [16]. 

### 2.4. Statistics and Software

Descriptive statistics were used to summarize findings on Excel software. Continuous variables are expressed as the median and range. Categorical variables are expressed as percentages and proportions. Resistance rates to antibiotics were calculated by using Whonet software.

Individual patients’ written informed consent for the data collection was obtained at admission. The hospital’s Institutional Review Board gave its approval for the study (25322/12.10.2018).

## 3. Results

The number of hospitalized patients decreased during the five years evaluated, but the number of patients treated with antibiotics was relatively stationary. Worth noting is the slight decrease in the percentage of patients who were administered antibiotics in 2020, but a significant increase in the number of patients treated with antibiotics in 2021, under the COVID-19 pandemic conditions (Table 1). 

Antibiotics were prescribed in 94,299 of the 175,272 patients (53.8%) which corresponds to a total number of DDDs/100 BD of 13,677.4 and about half of them were prescribed in a single year, 2021. Moreover, at the level of 2021, there was over a double of the consumption of antibiotics compared to 2020 or a tripling of it compared to the years 2017, 2018, and 2019.

Table 2 documents that the surgical wards had the highest rate of antibiotic prescriptions expressed as DDD/100 BD (65.42%), followed by medical wards (20.73%), and ICU (13.84%). The antibiotics were administered parenterally in 89.63% of the patients, and orally in the rest of the cases. Surgical and medical prophylaxis (67.45%) was the most prevalent reason for the administration of antibiotics.

The detailed pattern of antibiotic prescription is shown in Table 3. Most prescribed antibiotics were ceftriaxone (26.46%), metronidazole (13.05%), cefuroxime (10.96%), ampicillin (6.07%), ciprofloxacin (5.17%), amikacin (4.64%), amoxicillin/clavulanic acid (4.04%), cefixime (3.51%), gentamicin (3.14%), amoxicillin (2.37%), cefoperazone/sulbactam (2.34%), meropenem (2.10%), and clindamycin (2.03%), all belonging to group J01 antibiotics for systemic use. Further, rifaximin, which belongs to group A07 with local intestinal action, was included in the group of the most used antibiotics. The first ten antibiotics prescribed represented 77.03% of the total, and the first three antibiotics represented 50.46%, i.e., half of the total amount of antibiotics administered in terms of the frequency of their prescription, in the period 2017–2021 being represented by ceftriaxone, metronidazole, and cefuroxime. In contrast, erythromycin, fosfomycin, ceftazidime/avibactam, cefazolin, and imipenem/cilastatin/relebactam were only prescribed in the last two/three evaluated years and in small amounts.

The evaluation of antibiotic consumption by class shows a high consumption of more than half (62.46%) of beta-lactams, followed by azoles, fluoroquinolones, and aminoglycosides in approximately equal percentages (7.96 vs. 7.78%). Rifamycins, lincosamides, glycopeptides, tetracyclines, oxazolidinones, and polymyxins were prescribed in percentages below 3% of the total consumption of antibiotics by class (Table 4).

Third generation cephalosporins were the most common antibiotics (32.46%), followed by second generation cephalosporins (11.05%). Besides cephalosporin prescriptions accounted for about 43.73% of all antibiotic prescriptions. Carbapenem and glycopeptides accounted for 2.48% and 1.42%, respectively, of the prescriptions.

Figure 1, Figure 2 and Figure 3 show the most prescribed antibiotics at the ward-level. The most prescribed antibiotic in surgical wards were ceftriaxone, accounting for 22.81%, metronidazole (16.38%), cefuroxime (14.79%), ampicillin (7.75%), amikacin (6.09%), cefixime (5.2%), gentamicin (4.04%), ciprofloxacin (3.95%), amoxicillin (3.31%), amoxicillin/clavulanic (2.66%), and clindamycin (2.65%). The least prescribed antibiotics were cefepime (0.10%), erythromycin (0.08%), azithromycin (0.08%), linezolid (0.07%), imipenem/cilastin (0.04%), cefazolin (0.03%), ceftazidime/avibactam (0.01%), and fosfomycin (0.01%).

Similarly, in the medical wards, the most prescribed were ceftriaxone (23.98%), rifaximin (12.66%), ciprofloxacin (9.52%), metronidazole (7.34%), amoxicillin/clavulanic acid (6.55%), cefoperazone/sulbactam (5.70%), ampicillin (4.24%), meropenem (3.88%), ceftazidime (3.30%), cefuroxime (3.19%), cefoperazone (2.52%), and amikacin (1.98%), while ceftazidime/avibactam (0.10%), imipenem/cilastin (0.09%), ertapenem (0.05%), cefaclor (0.04%), ofloxacin (0.03%), rifampicin (0.02%), and erythromycin (0.02%) were less prescribed.

**Figure 2 healthcare-11-01302-f002:**
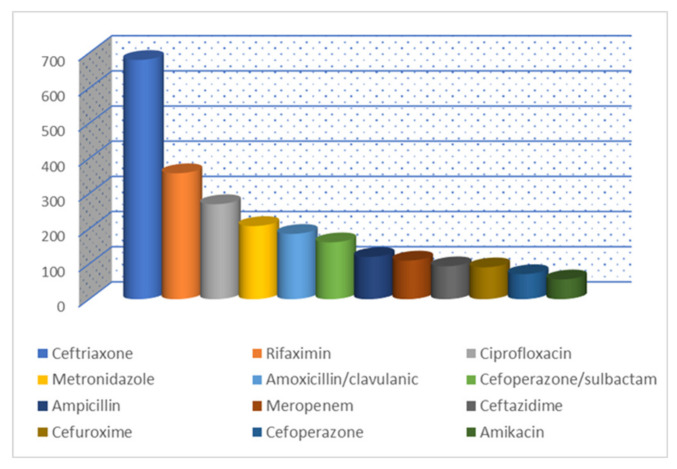
The most prescribed antibiotic at medical wards between 2017–2021 (cumulative report).

The same ceftriaxone was the most prescribed in the ICU (47.78%), followed by amoxicillin/clavulanic acid (6.83%), metronidazole (5.71%), cefoperazone/sulbactam (5.49%), ciprofloxacin (4.44%), cefuroxime (4.43%), meropenem (3.59%), moxifloxacin (2.71%), levofloxacin (2.01%), vancomycin (1.82%) amikacin (1.76%), colistin (1.47%), and gentamicin (1.46%). The least prescribed antibiotics were clarithromycin (0.17%), fosfomycin (0.16%), amoxicillin (0.08%), norfloxacin (0.08%), azithromycin (0.07%), cefixime (0.06%), cefaclor (0.03%), rifampicin (0.03%), cefazolin (0.01%), and ofloxacin (0.01%).

**Figure 3 healthcare-11-01302-f003:**
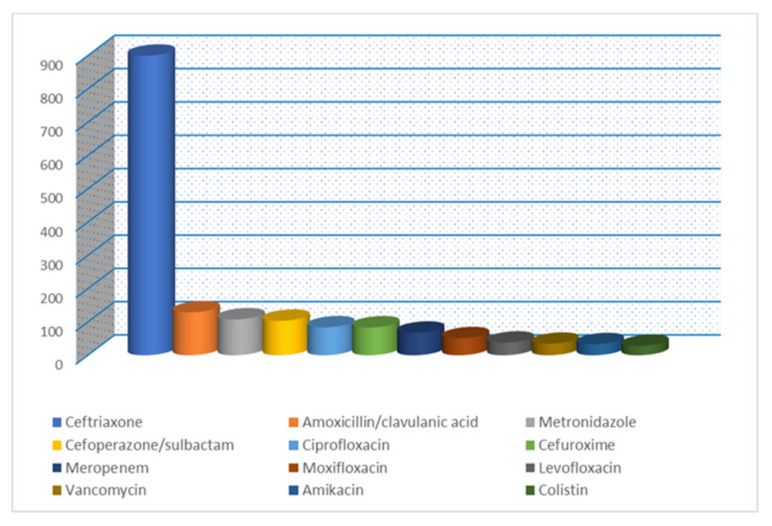
The most prescribed antibiotic at ICU between 2017–2021 (cumulative report).

According to AWaRe classification, 59.69% of the antibiotics prescribed were in the Watch Group class while only 37.07 were in the Access Group regardless of the evaluated year. Not Recommended Group class antibiotics were more highly used than reserve one and the evolution during the five evaluated years does not show a trend in decreasing the prescription of antibiotics from these classes. It should be mentioned that only cefoperazone/sulbactam was prescribed from the Not Recommended Group in our hospital (Table 5).

Comparative evaluation of the consumption of antibiotics according to AWaRe classification showed that most Access Group class antibiotics were prescribed in surgical wards, but without reaching the percentage recommended by WHO. A worrying aspect was the approximately equal percentage regarding the prescription of antibiotics from the class Not Recommended in medical wards and the ICU at 5.70% and 5.49%, respectively. Antibiotics from the Reserve Group were especially prescribed in the ICU (Figure 4).

We set out to analyze the resistance rates of the prescribed antibiotics. As Table 6 shows, minimal variations in antibiotic resistance were recorded when we compared them during the five years. Increasing rates of resistance showed ampicillin, amoxicillin/clavulanic acid, cefuroxime, meropenem, ceftazidime, levofloxacin, vancomycin, oxacillin, piperacillin/tazobactam, ampicillin/sulbactam, azithromycin, erythromycin, and ceftazidime/avibactam. Increased and relatively constant rates of resistance around 50% were presented by ceftriaxone, and ciprofloxacin. Cefixime, penicillin G, and oxacillin presented the highest resistance rates at around 60%. Resistance rates remained relatively low for amikacin, teicoplanin, colistin, rifampin, and linezolid.

## 4. Discussion

Since antibiotics were discovered, they have saved lives in patients with infectious diseases, but at the same time the development of antimicrobial resistance has occurred. Efforts to develop new therapeutic solutions against infectious agents, including herbal ones, have led to modest results [21,22].

Our study demonstrates a high prevalence of antibiotic use in patients from our hospital for five years. More than 50% of our patients received antibiotics, the same was reported by Castro Nunes et al. from a European country [23], but less than reported in a study conducted in Ghana [24]. A study conducted by Franchi et al. showed a prevalence of patients prescribed with antibiotics at about 33% and it seems to remain constant, and 46% for another study conducted by Omulo et al. [25,26]. In our study there is a marked increase in the consumption of antibiotics at the level of 2021. During the COVID-19 pandemic, in some studies, the use of antibiotics increased due to secondary bacterial infections, but we have not explored this aspect yet. Some studies showed a constant or a slight increase antibiotic use during COVID-19 pandemic contrary with our results [27]. In the same context, the consumption of antibiotics in the ICU remained relatively flat and increased in the surgical and medical wards. In other studies, prescription of antibiotics in the ICU decreased during the COVID-19 pandemic [28]. These results indicate differences in the management of COVID-19 patients and application and respecting of standards to ensure judicious use of antibiotics across health facilities. 

The evaluation in this study included antibiotic use for prophylaxis and therapeutical purposes. The prevalence of antibiotic use was concentrated especially in the surgical wards where one third of the total amount of antibiotic use was for surgical prophylaxis (60.23%) If wards are compared, the surgical ward’s rate of antibiotic prescriptions was the highest in the entire hospital (65.42%). This result could be explained by prophylactic hospital policies, but it is in concordance to other similar studies where ceftriaxone and metronidazole were the most prescribed for this purpose [29].

The parenteral route was the most used (89.63%) similar to other studies. This is a result of the type of hospital (emergency one), late presentation of critically ill patients, comorbidities, advanced age, and difficulties for oral intake. In addition, the most prescribed antibiotics in our study (cephalosporins) are available as parenteral injections. A reduced parenteral route and increase in the oral route of administration could be another strategy to reduce cost of healthcare by shortening of hospitalization and reducing the risk of catheter-associated infections.

Ceftriaxone, metronidazole, and cefuroxime were the most prescribed. Prescribing of broad-spectrum antibiotics is the first option for a large number of prescribers reported in many other studies, but this practice is responsible for promoting an emergency of antibiotic resistance [30]. The prescription of broad-spectrum antibiotics is not necessarily a decision for a successful treatment, even in the case of limited microbiology facilities. In one study, broad-spectrum antibiotics were associated with increased mortality and other poor outcomes in community-onset pneumonia [31].

We also reported an increased trend to prescribe beta-lactams during the evaluated period, especially in 2021. This increase could be explained by the difficulty in distinguishing between viral and bacterial etiology in some critically ill patients or uncertainty in treating COVID-19 patients. A study about sales of broad-spectrum antibiotic in 71 countries demonstrated an increase in the middle of the 2020 COVID-19 pandemic year especially for four broad-spectrum antibiotics: cephalosporins, penicillins, macrolides, and tetracyclines [32].

There were differences in the prescriptions between our wards but ceftriaxone was the most prescribed in all of them, accounting for 26.46% of prescriptions. Otherwise, third generation cephalosporins were among the most prescribed antibiotics (32.36%) and the prescribing rate was less than seen in other published studies [33,34,35]. On the other hand, during the COVID-19 pandemic in a tertiary hospital in Israel, a continuous decrease in antibiotic consumption was observed, and third generation cephalosporins were the most prescribed at more than a half [36].

The easier and most useful to support antibiotic stewardship is AWaRe classification provided by the WHO. Antibiotics are grouped into three major classes and a not recommended one. The WHO encourages a priority use for the Access Group and recommends that at least 60% of the overall antibiotic use should belong to this group. Our study identified that Watch Group antibiotics were more frequently prescribed (59.69%) contrary to this recommendation. The Watch Groups consists mostly of broad-spectrum antibiotics, those antibiotics more frequently responsible for infections with multidrug resistant pathogens and microbiota dysbiosis.

Access Group antibiotics are generally narrow-spectrum antibiotics, but they were prescribed only in 37.07% of patients, especially at the surgical wards. Even if antibiotics from the Reserve Group were prescribed only in the ICU and in relatively small quantities (0.82%), a problem identified in this study was the prescription of Not Recommended antibiotics in a similar proportion in ICU and medical wards. 

The consumption of antibiotics evaluated at the same time as the sensitivity and resistance rates confirms an otherwise well-known aspect: that a high consumption of a certain antibiotic leads to an increase in the resistance rate and could indicate which intervention is necessary.

There are limitations in this study. First, the evaluation was limited to a single hospital, but with many specialties including acute and chronic diseases and ICU. This also explains the wide range of antibiotics prescribed. Our data may not be representative of the large hospitals in Romania that would be our study population of interest. Second, we explored an annual assessment without observing monthly variations. Third, the evaluation did not consider the duration of the antibiotic prescriptions. More accurate results will be known only after the evaluation of optimal duration antibiotic prescription, as an evaluation criterion. Next, we expressed antibiotic consumption by DDD instead of days of therapy (DOT). According to a recent guideline for antibiotic stewardship programs, DOT is preferred to DDD as a measure of antibiotic consumption [37]. However, we could not use DOT because only the total amount of antibiotic consumption per patient was available. In addition, results of our study show the possibility of using only a few antibiotics or better classes to track the total antibiotic consumption at the hospital level. Our results are based on medical reports from the pharmacy but specific data on the antibiotic indication, diagnostic, and treatment duration are clearly insufficient as defined in the patient’s medical record. We did not evaluate antibiotic supply data in the hospital, which could influence the pattern of prescribing. 

## 5. Conclusions

We demonstrated the pattern of antibiotics prescribed in our patients despite measures provided by the antimicrobial stewardship program. Considering that our hospital is an emergency one, it is estimated that the antibiotic prescription is higher and intervention is required. A significant proportion of antibiotic prescriptions were cephalosporins (third and second generation) followed by azoles, fluoroquinolones, and aminoglycosides. In our study, antibiotic prescribing patterns differed from those indicated by WHO standard treatment guidelines. Most of these were in the form of parenteral drug formulations and belong to Watch Group antibiotics. 

Our results provide key areas to future quality improvement measures and capacity in the development of reliable ASPs in our hospital. Targeted AMS interventions are required first to reduce prescriptions. Next is the evaluation of appropriateness of antibiotic prescriptions, medical and patient education, and antimicrobial surveillance. Implementing the WHO AWaRe metrics is also recommended to manage antibiotic prescriptions and develop guidelines.

## Figures and Tables

**Figure 1 healthcare-11-01302-f001:**
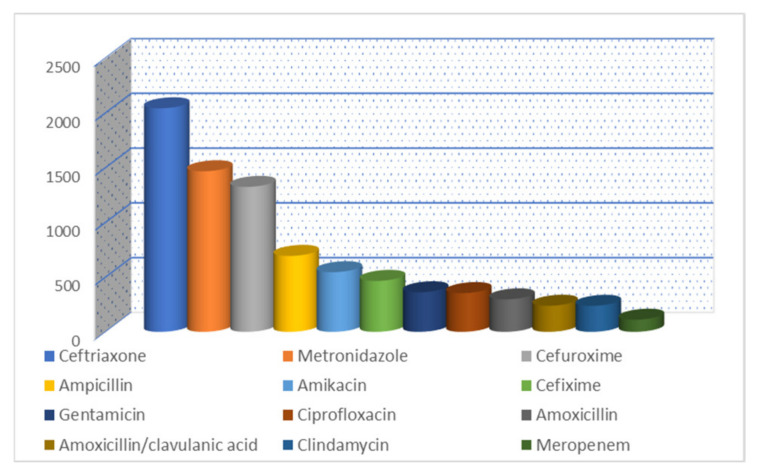
The most prescribed antibiotic at surgical wards between 2017–2021 (cumulative report).

**Figure 4 healthcare-11-01302-f004:**
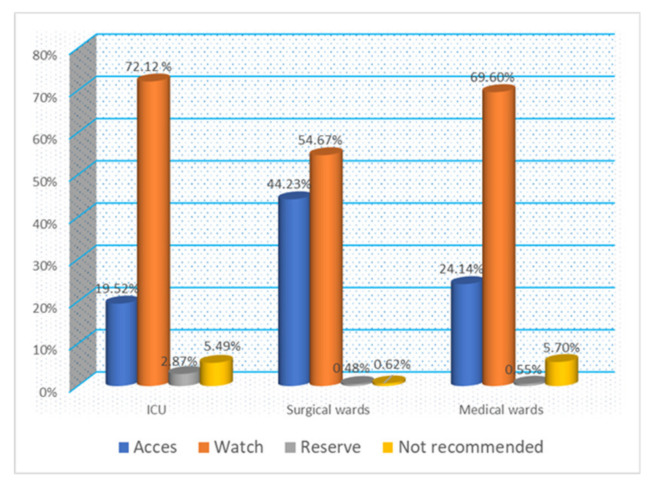
Antibiotics prescription according to WHO AWaRe classification by wards between 2017–2021 (cumulative report).

**Table 1 healthcare-11-01302-t001:** Total number of inpatients treated with antibiotics.

	2017	2018	2019	2020	2021	Total
Number of patients	41,598	40,038	39,743	26,138	27,755	175,272
Number of patients treated with antibiotics	22,928	21,881	21,182	13,135	15 173	94,299
%	55.12	54.65	53.30	50.25	54.67	53.80

**Table 2 healthcare-11-01302-t002:** Overall antibiotic use expressed as DDD/100 BD.

	2017	2018	2019	2020	2021	Total
Intensive care unit	398.87	368.77	381.04	389.66	344.59	1882.93
Surgical wards	1163.48	871.06	933.13	1559.36	4421.46	8948.49
Medical wards	415.91	421.38	350.21	432.75	1215.75	2836
Total	1978.26	1661.21	1664.38	2381.77	5981.80	13,677.42

**Table 3 healthcare-11-01302-t003:** Antibiotics prescribed by substance name expressed as DDD/100 BD.

ATC Code	Substance Name	2017	2018	2019	2020	2021	Total
J01DD04	Ceftriaxone	613.94	584.05	450.90	531.23	1438.52	3618.64
J01XD01	Metronidazole	110.79	43.39	104.81	298.63	1226.74	1784.37
J01DC02	Cefuroxime	167.68	198.96	255.52	427.52	449.36	1499.04
J01CA01	Ampicillin	129.11	133.26	132.46	138.85	297.06	830.74
J01MA02	Ciprofloxacin	109.29	105.78	88.42	115.67	288.43	707.60
J01GB06	Amikacin	217.12	37.46	34.24	36.45	308.83	634.11
J01CR02	Amoxicillin/clavulanic acid	106.56	78.07	61.47	94.83	211.72	552.64
J01DD08	Cefixime	9.26	9.83	9.95	16.08	434.70	479.83
J01GB03	Gentamicin	72.49	54.39	89.45	97.21	116.54	430.09
A07AA11	Rifaximin	73.57	77.45	67.63	66.89	123.53	409.06
J01CA04	Amoxicillin	38.49	43.22	43.03	28.36	170.93	324.04
J01DD62	Cefoperazone/sulbactam	36.83	46.77	34.06	56.09	145.68	319.43
J01DH02	Meropenem	16.23	22.04	31.90	71.97	145.61	287.74
J01FF01	Clindamycin	44.33	56.12	42.85	45.09	89.07	277.46
J01DD02	Ceftazidime	39.58	20.76	25.21	34.00	59.86	179.40
J01DD12	Cefoperazone	45.17	20.94	21.57	17.92	55,75	161.36
J01MA12	Levofloxacin	21.78	20.80	10.54	31.67	59.81	144.60
J01XA01	Vancomicin	8.67	11.02	18.22	39.13	57.63	134.66
J01MA14	Moxifloxacin	10.29	17.31	15.85	29.74	35.20	108.39
J01MA06	Norfloxacin	5.82	5.51	3.17	80.50	7.07	102.06
J01AA02	Doxycycline	2.91	3.22	8.76	10.23	54.61	79.72
J01CE01	Benzylpenicillin	29.39	5.65	22.71	13.99	2.38	74.13
J01XA02	Teicoplanin	1.86	1.93	22.20	13.32	20.06	59.38
J01FA09	Clarithromycin	5.54	6.03	7.77	7.85	31.24	58.42
J01XB01	Colistin	6.80	10.03	8.98	7.15	21.54	54.49
J01CF04	Oxacilin	11.14	6.75	9.78	12.50	12.13	52.30
J01DH03	Ertapenem	5.08	4.09	4.81	6.20	18.11	38.29
J04AB02	Rifampicin	3.39	0.29	4.46	7.74	15.92	31.79
J01CR05	Piperacillin/tazobactam	8.08	8.67	3.36	0.96	10.29	31.36
J01CR01	Ampicillin/sulbactam	2.84	5.96	5.33	3.80	11.71	29.65
J01DE01	Cefepime	0.00	0.27	2.34	5.88	19.09	27.57
J01MA01	Ofloxacin	15.70	4.47	2.31	0.68	3.89	27.06
J01AA12	Tigecycline	0.38	1.61	4.10	7.25	12.39	25.73
J01XX08	Linezolid	1.76	7.62	5.35	1.02	6.75	22.49
J01FA10	Azithromycin	0.63	0.39	0.25	14.08	4.87	20.21
J01DC04	Cefaclor	4.97	5.55	1.67	0.77	0	12.96
J01DH51	Imipenem/Cilastin	0.79	1.57	3.04	2.14	4.93	12.47
J01FA01	Erythromycin	0	0	2.60	1.92	7.13	11.65
J01XX01	Fosfomycin	0	0	2.12	2.51	5.93	10.56
J01DD52	Ceftazidime/avibactam	0	0	0.83	1.65	6.92	9.40
J01DB04	Cefazolin	0	0	0.37	2.32	0	2.69
J01DH56	Imipenem/cilastatin/ relebactam	0	0	0	0	0.23	0.23
	Total	1978.26	1661.21	1664.38	2381.77	5981.80	13,677.4

**Table 4 healthcare-11-01302-t004:** Antibiotics prescription by pe classes expressed as DDD/100 BD.

	2017	2018	2019	2020	2021	Total
Penicillin’s	208.13	188.89	207.98	193.70	482.51	1281.20
Combination (beta-lactam/beta-lactamase inhibitors)	154.31	139.48	105.05	157.32	386.32	942.48
1st generation cephalosporins	-	-	0.37	2.32	-	2.69
2nd-generation cephalosporins	172.65	204.51	257.19	428.29	449.36	1511.99
3rd-generation cephalosporins	707.96	635.58	507.64	599.22	1988.83	4439.22
4th-generation cephalosporins	-	0.27	2.34	5.88	19.09	27.57
Carbapenems	22.10	27.70	39.75	80.31	168.88	338.73
Beta-lactams	1265.16	1196.41	1120.31	1467.04	3494.98	8543.90
Macrolides	6.16	6.42	10.62	23.84	43.24	90.28
Lincosamide	44.33	56.12	42.85	45.09	89.07	277.46
Aminoglycoside	289.62	91.85	123.69	133.67	425.38	1064.20
Glycopeptides	10.53	12.95	40.42	52.45	77.69	194.04
Polymyxins	6.80	10.03	8.98	7.15	21.54	54.49
Fluoroquinolones	162.89	153.86	120.29	258.26	394.40	1 089.70
Tetracyclines	3.29	4.82	12.86	17.48	67.00	105.45
Rifamycins	76.95	77.74	72.09	74.63	139.45	440.86
Azoles	110.79	43.39	104.81	298.63	1226.74	1784.37
Oxazolidinone	1.76	7.62	5.35	1.02	6.75	22.49
Other antibiotics for systemic use	-	-	2.12	2.51	5.93	10.56

**Table 5 healthcare-11-01302-t005:** Antibiotics prescription according to WHO AWaRe classification.

Group Antibiotics	2017	2018	2019	2020	2021	Total
Access	765.18	467.51	555.26	782.26	2501.74	5071.94 (37.07%)
Watch	1167.32	1127.69	1055.81	1526.35	3296.92	8174.10 (59.69%)
Reserve	8.94	19.25	19.26	17.07	47.83	112.35 (0.82%)
Not Recommended	36.83	46.77	34.06	56.09	145.68	319.43 (2.34%)

**Table 6 healthcare-11-01302-t006:** Resistance rates of prescribed antibiotics.

Antibiotic	% Resistance (%R 95%C.I.)					Coefficient of Variation
	2017	2018	2019	2020	2021	
Ceftriaxone	52.9 (50.7–55.3)	48.6 (40.2–55.3)	40.9 (37.3–44.2)	54.5 (51.0–58.1)	48.8 (45.0–53.1)	0.11
Cefuroxime	40.5 (37.4–43.7)	39.5 (37.1–42.2)	40.1 (36.4–43.9)	38.2 (34.3–42.3)	42.6 (38.7–48.4)	0.03
Ampicillin	49.1 (46.3–53.1)	79.8 (71.9–81.2)	68.3 (63.7–69.5)	65.1 (63.2–67.4)	69.4 (67.8–71.4)	0.17
Ciprofloxacin	53.6 (51.4–55.8)	52.9 (51.3–54.5)	48.4 (46.9–49.9)	46.8 (45.2–48.4)	54.7 (53.2–56.2)	0.07
Amikacin	14.9 (13.4–16.6)	12.9 (11.5–14.2)	13.7 (12.6–14.9)	11.4 (10.2–12.7)	18.9 (17.6–20.3)	0.20
Amoxicillin/ Clavulanic acid	26.6 (30.0–34.7)	61.7 (59.4–63.9)	51.9 (51.0–55.0)	48.6 (47.0–51.2)	52 (50.7–54.7)	0.27
Cefixime	53.6 (49.8–57.4)	52 (48.3–55.2)	54.9 (48.8–60.8)	61.6 (54.6–68.1)	62.5 (57.0–67.7)	0.08
Gentamicin	36.1 (33.6–38.7)	29.6 (28.1–31.1)	32.2 (30.8–33.6)	26.7 (25.2–28.3)	30.8 (29.4–32.2)	0.11
Meropenem	23.1 (20.5–25.9)	24.7 (23.1–26.4)	22.9 (21.6–24.3)	24.1 (22.5–25.7)	36.6 (35.0–38.2)	0.22
Clindamycin	42.9 (39.1–46.8)	58.8 (54.9–62.6)	42.6 (39.5–45.8)	44.7 (41.0–48.5)	35 (31.4–38.7)	0.19
Ceftazidime	38.2 (28.0–39.7)	40.5 (38.7–42.4)	39 (37.2–40.9)	33.2 (31.2–35.3)	42.6 (40.7–44.5)	0.09
Levofloxacin	40 (37.7–42.4)	42 (39.7–46.1)	45.8 (42.2–49.4)	46.9 (43.0–50.8)	53.1 (49.6–56.7)	0.11
Vancomycin	2.1 (1.8–4.2)	11.7 (10.2–14.3)	3 (2.2–10.0	3 (2.6–11.1)	10 (9.0–12.6)	0.76
Moxifloxacin	73.8 (67.7–79.1)	19.7 (16.7–23.0)	17.9 (15.5–20.8)	21 (16.9–23.3)	23.2 (20.3–26.2)	0.19
Penicillin G	63 (61.3–68.0)	62 (59.1–68.0)	74 (76.0–94.4)	58 (51.1–69.2)	63.6 (62.4–70.5)	0.11
Teicoplanin	1.7 (1.2–2.2)	7.9 (6.3–9.9)	7 (5.6–8.7)	6 (4.4–8.1)	5.8 (4.4–7.6)	0.42
Colistin	-	21 (17.9–24.5)	27.4 (24.2–30.8)	21.3 (17.9–25.2)	18.6 (16.0–21.5)	0.17
Oxacillin	4.3 (2.4–7.4)	76.1 (72.4–79.4)	64 (60.0–68.2)	63.6 (58.1–68.7)	66.9 (61.4–72.0)	0.17
Ertapenem	-	12.1 (10.7–13.7)	15.2 (13.6–16.9)	11.3 (9.8–13.0)	22.2 (20.4–24.1)	0.33
Rifampin	-	19.1 (16.1–22.5)	12.2 (9.5–15.5)	9.2 (6.5–12.9)	10.8 (7.7–14.7)	0.34
Piperacillin/ Tazobactam	18.5 (16.5–20.6)	36.7 (34.9–38.6)	31.5 (29.9–33.9)	37.5 (35.8–39.2)	43.5 (41.9–45.1)	0.28
Ampicillin/ Sulbactam	21.8 (19.3–24.5)	49 (41.31–52.9)	50.3 (42.3–58.3)	45.3 (39.9–50.8)	53.8 (50.0–57.6)	0.10
Cefepime	39.1 (34.1–44.3)	30 (28.3–31.8)	30.1 (28.5–31.)	26.9 (25.1–28.7)	35.1 (33.6–36.8)	0.15
Ofloxacin	31.9 (29.6–34.3)	36 (31.2–42.6)	44 (39.2–48.9)	40.1 (32.8–47.9)	29.6 (23.5–36.5)	0.16
Linezolid	-	6.8 (5.4–8.7)	6.8 (5.8–9.4)	4.2 (3.0–6.4)	4.5 (5.3–9.4)	0.25
Azithromycin	59.5 (51.6–66.9)	60.1 (54.6–65.1)	50 (34.8–65.2)	73.9 (51.3–88.9)	66.7 (44.7–83.6)	0.14
Imipenem/ Cilastin	36.1 (30.4–42.2)	27.6 (25.9–29.3)	27.4 (25.8–29.0)	30.3 (28.5–32.2)	37.1 (35.4–38.8)	0.15
Erythromycin	49 (44.7–53.3)	68.7 (65.6–71.6)	60.4 (57.9–62.9)	61.8 (58.9–64.7)	58.1 (54.9–61.2)	0.12
Ceftazidime/ Avibactam	-	-	28.1 (21.6–35.7)	32.1 (28.1–36.4)	53.2 (50.2–56.2)	0.36

## Data Availability

All the relevant data have been included in this study.

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
