# Peer review of "Pattern of Antibiotic Use among Hospitalized Patients at a Level One Multidisciplinary Care Hospital"

_healthcare, 2023, doi:10.3390/healthcare11091302_

Round 1
Reviewer 1 Report
Despite the results being based on pharmacy medical reports, where specific data on antibiotic indication, diagnosis and duration of treatment are insufficient, the present study presents important data that deserve to be published.
1- The study demonstrates a high prevalence of antibiotic use in patients from Emergency Clinical County Hospital of Oradea, Romania for five years.
2- Data showed an increase of antibiotic consumption in 2021, while other studies showed a constant or a slight increase in antibiotic use during the COVID-19 pandemic, indicating significant differences in the management of patients with COVID-19 in health facilities.
3- The study identified that antibiotics from the surveillance group were frequently prescribed contrary to the WHO AWaRe classification.
Author Response
Despite the results being based on pharmacy medical reports, where specific data on antibiotic indication, diagnosis and duration of treatment are insufficient, the present study presents important data that deserve to be published.
1- The study demonstrates a high prevalence of antibiotic use in patients from Emergency Clinical County Hospital of Oradea, Romania for five years.
2- Data showed an increase of antibiotic consumption in 2021, while other studies showed a constant or a slight increase in antibiotic use during the COVID-19 pandemic, indicating significant differences in the management of patients with COVID-19 in health facilities.
3- The study identified that antibiotics from the surveillance group were frequently prescribed contrary to the WHO AWaRe classification.
Thank you for the valuable comments, otherwise the main conclusions of the study. These aspects show the need for rigorous control with short and long term effects. As suggested one of the reviewers, we have added data on the resistance rates of tested and prescribed antibiotics.
Reviewer 2 Report
The manuscript of Hodosan et al. describes a retrospective investigation of the pattern of antibiotic use among patients hospitalized at level one multidisciplinary care hospital (Romania) during 5 years (2017 to 2021). The major conclusions were that the use of antibiotics was high in all hospital wards (59.69%) and the most prescribed antibiotic class was cephalosporin (43.73%). The prescriction of antibiotics according to WHOs standard treatment guidelines was not followed and most of the antibiotics administered were from the Watch group. During the 5 years this procedure was constant and no improvement was observed. Therefore, this work is important to avoid antibiotic misuse, and understand and improve the antibiotic prescription procedure according to the WHO’s AWaRe classification.
Major comments:
1) Table 2: Please confirm if the Total/Total value is 13677,42 or 13667,42. And review the percentages of antibiotic prescription accordingly (Lines 180 to 181).
2) Table3: Please review also the total value in Table 3. And review the percentages of each antibiotic prescription accordingly (Lines 186 to 192).
3) Page 5, lines 195 to 197: Please review the phrase. In the case of cefaclor the affirmation is not correct, because it is shown a decrease of the use of the antibiotic from 2017 to 2021.
4) Table 4: In the text, lines 155 to 156, it is described two groups beta-lactam/beta-lactamase inhibitors and cephalosporins/ beta-lactamase inhibitors, however in the table only appears beta-lactam/beta-lactamase inhibitors group. Please review text or table accordingly.
5) Table 4: Add the total value of Beta lactams, per year and the sum of the 5 years.
6) Table 4: Please review the Total value of 1st generation cephalosporins.
Minor comments:
1) Page 2, line55-56: Please review the phrase.
2) The species name after appearing the first time, can be abbreviated. Example: Escherichia coli can be E. coli, Klebsiella pneumoniae ban be K. pneumoniae.
3) Page 2, line 48 and Page 3, line 114: The Abbreviation WHO can be used solely after appearing the first time described.
4) Table 3: Please correct on the table gentamycin to gentamicin and rifaxime to rifaximin.
5) Figure 1: Please increase the quality and size of the figure.
Author Response
The manuscript of Hodosan et al. describes a retrospective investigation of the pattern of antibiotic use among patients hospitalized at level one multidisciplinary care hospital (Romania) during 5 years (2017 to 2021). The major conclusions were that the use of antibiotics was high in all hospital wards (59.69%) and the most prescribed antibiotic class was cephalosporin (43.73%). The prescriction of antibiotics according to WHOs standard treatment guidelines was not followed and most of the antibiotics administered were from the Watch group. During the 5 years this procedure was constant and no improvement was observed. Therefore, this work is important to avoid antibiotic misuse, and understand and improve the antibiotic prescription procedure according to the WHO’s AWaRe classification.
Major comments:
1) Table 2: Please confirm if the Total/Total value is 13677,42 or 13667,42. And review the percentages of antibiotic prescription accordingly (Lines 180 to 181). Thank you for your observation, the total number of DDD/100 BD is 13677,42.
2) Table3: Please review also the total value in Table 3. And review the percentages of each antibiotic prescription accordingly (Lines 186 to 192). We checked the percentages being reported to the same value 13 677.42
3) Page 5, lines 195 to 197: Please review the phrase. In the case of cefaclor the affirmation is not correct, because it is shown a decrease of the use of the antibiotic from 2017 to 2021. Thank you for your observation, we rephrased.
4) Table 4: In the text, lines 155 to 156, it is described two groups beta-lactam/beta-lactamase inhibitors and cephalosporins/ beta-lactamase inhibitors, however in the table only appears beta-lactam/beta-lactamase inhibitors group. Please review text or table accordingly. Thank you, we added cephalosporins/ beta-lactamase inhibitors, they are included in this category.
5) Table 4: Add the total value of Beta lactams, per year and the sum of the 5 years. Thank you, we added.
6) Table 4: Please review the Total value of 1st generation cephalosporins. Thank you, we corrected.
Minor comments:
1) Page 2, line55-56: Please review the phrase.
2) The species name after appearing the first time, can be abbreviated. Example: Escherichia coli can be E. coli, Klebsiella pneumoniae ban be K. pneumoniae. Thank you, we corrected
3) Page 2, line 48 and Page 3, line 114: The Abbreviation WHO can be used solely after appearing the first time described. Thank you, we corrected
4) Table 3: Please correct on the table gentamycin to gentamicin and rifaxime to rifaximin. Thank you, we corrected
5) Figure 1: Please increase the quality and size of the figure. Thank you, we changed them all.
As suggested one of the reviewers, we have added data on the resistance rates of tested and prescribed antibiotics.
Reviewer 3 Report
line 156: 1st generation written as one word
line 169-170: seem very categorical, is there any evidence for this?
The use of decimal point system is not clear in lines 173-174. This occurs at several places in the document
In lines 185-190, authors indicate most prescribed...... at the start of sentence but report several values ranging from 26.46% to 2.03%
Table 4 would have been better appreciated as a graph to monitor trends over the period.
There's a typographical error in line 284, ... for o large ....
Author Response
line 156: 1st generation written as one word Thank you, we corrected
line 169-170: seem very categorical, is there any evidence for this? Thank you, we corrected
The use of decimal point system is not clear in lines 173-174. This occurs at several places in the document Thank you, we corrected
In lines 185-190, authors indicate most prescribed...... at the start of sentence but report several values ranging from 26.46% to 2.03%. As can be seen from table 3, these are the most prescribed in descending order expressed as a percentage of the total number of doses per 100 days of hospitalization.
Table 4 would have been better appreciated as a graph to monitor trends over the period. We corrected the table according to the recommendations of another reviewer, but we can changed into a figure.
There's a typographical error in line 284, ... for o large .... Thank you, we corrected
As suggested one of the reviewers, we have added data on the resistance rates of tested and prescribed antibiotics.
Reviewer 4 Report
The current review “Pattern of antibiotic use among hospitalized patients at a level one multidisciplinary care hospital for five years” help to understand antibiotic prescription to identify the possible measure and intervention for promoting the rational use of antibiotics by using WHO’s Access, Watch, Reserve (AWaRe) classification and or to plan future antimicrobial stewardship efforts. However, several issues must be solved by emphasizing the following points before this work may be accepted.
In the Introduction section:
Does Antimicrobial resistance differ from antibiotic microbial resistance? If not, then please abbreviate them as AMR. Similarly, for antibiotic stewardship programs in line 74, please provide the acronym. Provide the supporting references for the data mentioned in paragraphs 3 and 8.
The materials and methods are short, sketchy, and difficult to understand. The authors should divide it into 3-4 subsections for better visualization. Does the author follow any inclusion/exclusion criteria for the extraction of information? Do the authors use any software for analysis? Provide the abbreviation for intensive care unit (line 180). In line 284, what does ‘o’ represent?
Author Response
The current review “Pattern of antibiotic use among hospitalized patients at a level one multidisciplinary care hospital for five years” help to understand antibiotic prescription to identify the possible measure and intervention for promoting the rational use of antibiotics by using WHO’s Access, Watch, Reserve (AWaRe) classification and or to plan future antimicrobial stewardship efforts. However, several issues must be solved by emphasizing the following points before this work may be accepted.
In the Introduction section:
Does Antimicrobial resistance differ from antibiotic microbial resistance? If not, then please abbreviate them as AMR. Similarly, for antibiotic stewardship programs in line 74, please provide the acronym. Thank you, we corrected
Provide the supporting references for the data mentioned in paragraphs 3 and 8. There are ref 5 for paragraph 3 and ref 16,17,18,19 for the paragraph 8
The materials and methods are short, sketchy, and difficult to understand. The authors should divide it into 3-4 subsections for better visualization. Does the author follow any inclusion/exclusion criteria for the extraction of information? Do the authors use any software for analysis? Provide the abbreviation for intensive care unit (line 180). In line 284, what does ‘o’ represent? Thank you, we followed your suggestion.
As suggested one of the reviewers, we have added data on the resistance rates of tested and prescribed antibiotics.
Reviewer 5 Report
Comment to the authors,
Line 38: antibiotic microbial resistance (AMR) should be replaced with "antimicrobial resistance (AMR)".
Line 66: multidrug-resistant should be replaced with "MDR".
Introduction: First write the full name of the bacteria in the text and then as below.
Klebsiella pneumoniae " K. pneumoniae " OR Staphylococcus aureus " S. aureus" AND ……
Line 107: Defined Daily Dose should be replaced with "DDD".
Line 114: World Health Organization’s (WHO) should be replaced with "WHO".
Line 156: 1stgeneration cephalosporins should be replaced with "1st-generation cephalosporins ".
Line 173: 94.299 should be replaced with "94 299".
Table 1-3: Please report the numbers in the table in the same way. 398.87 or 398 87.
Table 4: Some antibiotic names are in italics. Please correct it.
Fig1-3 are the total prescription statistics of antibiotics? Please indicate what period of time they are related to.
Have secondary bacterial infections increased in the ICU of your hospital during the covid pandemic compared to previous and subsequent years or not?
During the covid pandemic, in some studies, the use of antibiotics increased due to secondary bacterial infections. Please mention this point in the discussion section.
Line 272: (60.23%) When compared wards…..Before when is the sentence finished?
It is suggested that, if possible, due to the increase in the use of antibiotics such as ceftazidime, meropenem, colistin and tigecycline, the amount of antibiotic resistance should be reported in a table every year.
Author Response
Comment to the authors,
Line 38: antibiotic microbial resistance (AMR) should be replaced with "antimicrobial resistance (AMR)". Thank you, we corrected
Line 66: multidrug-resistant should be replaced with "MDR". Thank you, we corrected
Introduction: First write the full name of the bacteria in the text and then as below Klebsiella pneumoniae " K. pneumoniae " OR Staphylococcus aureus " S. aureus" AND ……. Thank you, we corrected
Line 107: Defined Daily Dose should be replaced with "DDD". Thank you, we corrected
Line 114: World Health Organization’s (WHO) should be replaced with "WHO". Thank you, we corrected
Line 156: 1stgeneration cephalosporins should be replaced with "1st-generation cephalosporins ". Thank you, we corrected
Line 173: 94.299 should be replaced with "94 299". Thank you, we corrected
Table 1-3: Please report the numbers in the table in the same way. 398.87 or 398 87. Thank you, we corrected
Table 4: Some antibiotic names are in italics. Please correct it. Thank you, we corrected
Fig1-3 are the total prescription statistics of antibiotics? Please indicate what period of time they are related to. Thank you, we corrected
Have secondary bacterial infections increased in the ICU of your hospital during the covid pandemic compared to previous and subsequent years or not? During the covid pandemic, in some studies, the use of antibiotics increased due to secondary bacterial infections. Please mention this point in the discussion section. Unfortunately, we are going to explore these aspects further.
Line 272: (60.23%) When compared wards…..Before when is the sentence finished? Thank you, we corrected
It is suggested that, if possible, due to the increase in the use of antibiotics such as ceftazidime, meropenem, colistin and tigecycline, the amount of antibiotic resistance should be reported in a table every year.
As suggested one of the reviewers, we have added data on the resistance rates of tested and prescribed antibiotics.
Round 2
Reviewer 4 Report
The authors have addressed all my comments in the revised manuscript.
Author Response
Dear Reviewer
we hope all the requirements to be solved.